# Hepatitis B Virus X Protein Stimulates Hepatitis C Virus (HCV) Replication by Protecting HCV Core Protein from E6AP-Mediated Proteasomal Degradation

Hyunyoung Yoon,[a] Jiwoo Han,[a] Kyung Lib Jang[a,b,c]

[a]Department of Integrated Biological Science, The Graduate School, Pusan National University, Busan, Republic of Korea
[b]Department of Microbiology, College of Natural Science, Pusan National University, Busan, Republic of Korea
[c]Microbiological Resource Research Institute, Pusan National University, Busan, Republic of Korea

**ABSTRACT** Most clinical and experimental studies have suggested that hepatitis C virus (HCV) is dominant over hepatitis B virus (HBV) during coinfection, although the underlying mechanism remains unclear. In this study, we found that the HBV X protein (HBx) upregulates the levels of the HCV core protein to stimulate HCV replication during coinfection in human hepatoma cells. For this purpose, HBx upregulated both the protein levels and enzyme activities of cellular DNA methyltransferase 1 (DNMT1) and DNMT3b, and this subsequently reduced the expression levels of the E6-associated protein (E6AP), an E3 ligase of the HCV core protein, via DNA methylation. The ubiquitin-dependent proteasomal degradation of the HCV core protein was severely impaired in the presence of HBx, whereas this effect was not observed when *E6AP* was either ectopically expressed or restored by treatment with 5-aza-2′dC or DNMT1 knockdown. The effect of HBx on the HCV core protein was accurately reproduced in HBV/HCV coinfection systems, which were established by either monoinfection by HCV in Huh7D cells transfected with a 1.2-mer HBV replicon or coinfection by HBV and HCV in Huh7D-Na$^+$-taurocholate cotransporting polypeptide cells, providing evidence for the stimulation of HCV replication by HBx. The present study may provide insights into understanding HCV dominance during HBV/HCV coinfection in patients.

**IMPORTANCE** Hepatitis B virus (HBV) and hepatitis C virus (HCV) are major human pathogens that cause a substantial proportion of liver diseases worldwide. As the two hepatotropic viruses have the same modes of transmission, coinfection is often observed, especially in areas and populations where HBV is endemic. High-risk populations include people who inject drugs. Both clinical and experimental studies have shown that HCV is more dominant than HBV during coinfection, but the underlying mechanism remains unclear. In this study, we show that HBV X protein (HBx) stimulates HCV replication by inhibiting the expression of E6-associated protein (*E6AP*) via DNA methylation, thereby protecting the HCV core protein from proteasomal degradation, which can contribute to HCV dominance during HBV/HCV coinfection.

**KEYWORDS** DNA methylation, E6-associated protein, HBx, HCV core protein, ubiquitin-proteasome system

Address correspondence to Kyung Lib Jang, kljang@pusan.ac.kr.

The authors declare no conflict of interest.

**H**epatitis B virus (HBV) and hepatitis C virus (HCV) are major human pathogens that infect the human liver and cause hepatitis, cirrhosis, and hepatocellular carcinoma (HCC) (1). HBV and HCV are phylogenetically unrelated and belong to the *Hepadnaviridae* and *Flaviviridae* families, respectively. However, they do have similar transmission routes, such as intravenous drug use and blood transfusion, which can lead to coinfection with the two viruses (2, 3). The estimated prevalences of HBV/HCV coinfection are 10 to 20% in patients with chronic HBV infection and 2 to 10% in patients with chronic HCV infection, with wide

variations depending on the geographical region, population, and study design (3–6). The coinfection rate can be over 40% in high-risk populations such as people who inject drugs (7). Coinfection with HBV and HCV is often associated with a higher risk of advanced liver diseases, including HCC, than with monoinfection (8, 9), indicating the importance of clinical and virological studies on HBV/HCV coinfection.

Both experimental and epidemiological studies have suggested that HCV is largely dominant over HBV during coinfection. For example, detectable levels of HCV viremia and extremely low titers of serum HBV DNA have been reported in patients (5, 6, 10). In cases of HCV superinfection, HBV e antigen seroconversion and HBV surface antigen (HBsAg) clearance have also been reported (11, 12). Similarly, acute HCV superinfection in chimpanzees with chronic HBV infection resulted in a significant reduction in serum HBsAg levels (13, 14). Moreover, the clearance of HCV with interferon alpha and ribavirin resulted in HBV reactivation in some coinfected patients (15, 16). Despite accumulating evidence for HCV dominance, its virological aspects and molecular mechanisms are poorly understood. Early studies using *in vitro* coinfection systems revealed that HCV dominance is mediated largely by HCV core proteins (17, 18). Subsequent studies have suggested several mechanisms of action of the HCV core protein, including the inhibition of HBV transcription and genome replication via direct interactions with HBV X protein (HBx) and HBV polymerase, respectively (19, 20). More recently, it has been demonstrated that HCV core protein inhibits HBV replication by downregulating HBx levels via Siah-1-mediated proteasomal degradation during coinfection (21). Most studies on HCV dominance so far have focused on the inhibition of HBV replication by the HCV core protein. However, little information is available on the regulation of HCV replication by HBV during coinfection.

Previous studies have demonstrated that the ubiquitin-proteasome system restricts HCV replication by controlling the amount of viral proteins, including the HCV core protein (22–24). E6-associated protein (E6AP) (also called ubiquitin protein ligase E3A [UBE3A]), which was first identified as a ubiquitin ligase that mediates the degradation of p53 in conjunction with the E6 protein of human papillomaviruses 16 and 18 (25, 26), induces the ubiquitination of HCV core protein, which subsequently leads to its proteasomal degradation (23). *E6AP* expression in the presence of HCV core protein appears to be regulated by DNA methylation at the CpG sites of the *E6AP* promoter (27). HBx is a well-known regulator of DNA methylation, which upregulates both protein and enzyme activity levels of cellular DNA methyltransferases (DNMTs) and silences tumor suppressor genes such as *CDKN2A* (p16), *CDKN1A* (p21), *CDH1* (E-cadherin), and *IGFBP3* via promoter hypermethylation (28–31). These observations prompted us to investigate whether HBx inhibits *E6AP* expression in the presence of HCV core protein via DNA methylation, prevents HCV core protein from E6AP-mediated ubiquitin-dependent proteasomal degradation, and stimulates HCV replication that can contribute to HCV dominance during coinfection.

## RESULTS

**HBV stimulates HCV replication during coinfection in cultured human hepatoma cells.** To investigate whether HBV affects HCV replication in cultured human hepatoma cells, Huh7D cells, which are highly susceptible to HCV infection (32), were first transfected with an HBV replicon, 1.2-mer WT (wild type), for 24 h and then infected with the cell culture-adapted HCV JFH1 strain for an additional 24 h, which mimicked superinfection by HCV in HBV-infected cells to establish coinfection with both viruses. In addition, Huh7D-Na$^+$-taurocholate cotransporting polypeptide (NTCP) cells, which can support the replication of HBV (33) and HCV (32), were simultaneously infected with HBV and HCV to establish coinfection. HBV replication originating from the 1.2-mer HBV replicon and HBV was confirmed by the detection of HBV proteins such as HBx and HBsAg (Fig. 1A and G; see also Fig. S1A at https://dv.pusan.ac.kr/SynapDocViewServer/viewer/doc.html?key=402882e382e741b101845c07289e5ac8&convType=html&convLocale=ko&contextPath=/SynapDocViewServer/), the detection of relaxed circular DNA (rcDNA) and covalently closed circular DNA (cccDNA) (see Fig. S1C at the URL mentioned above), and the subcellular localization of HBx (Fig. 1C and E; see also Fig. S1D at the URL mentioned above) in the infected cells. According to data from the immunofluorescence analysis (IFA), the HBV infection rates of the 1.2-mer HBV replicon and *in vitro* infection systems were

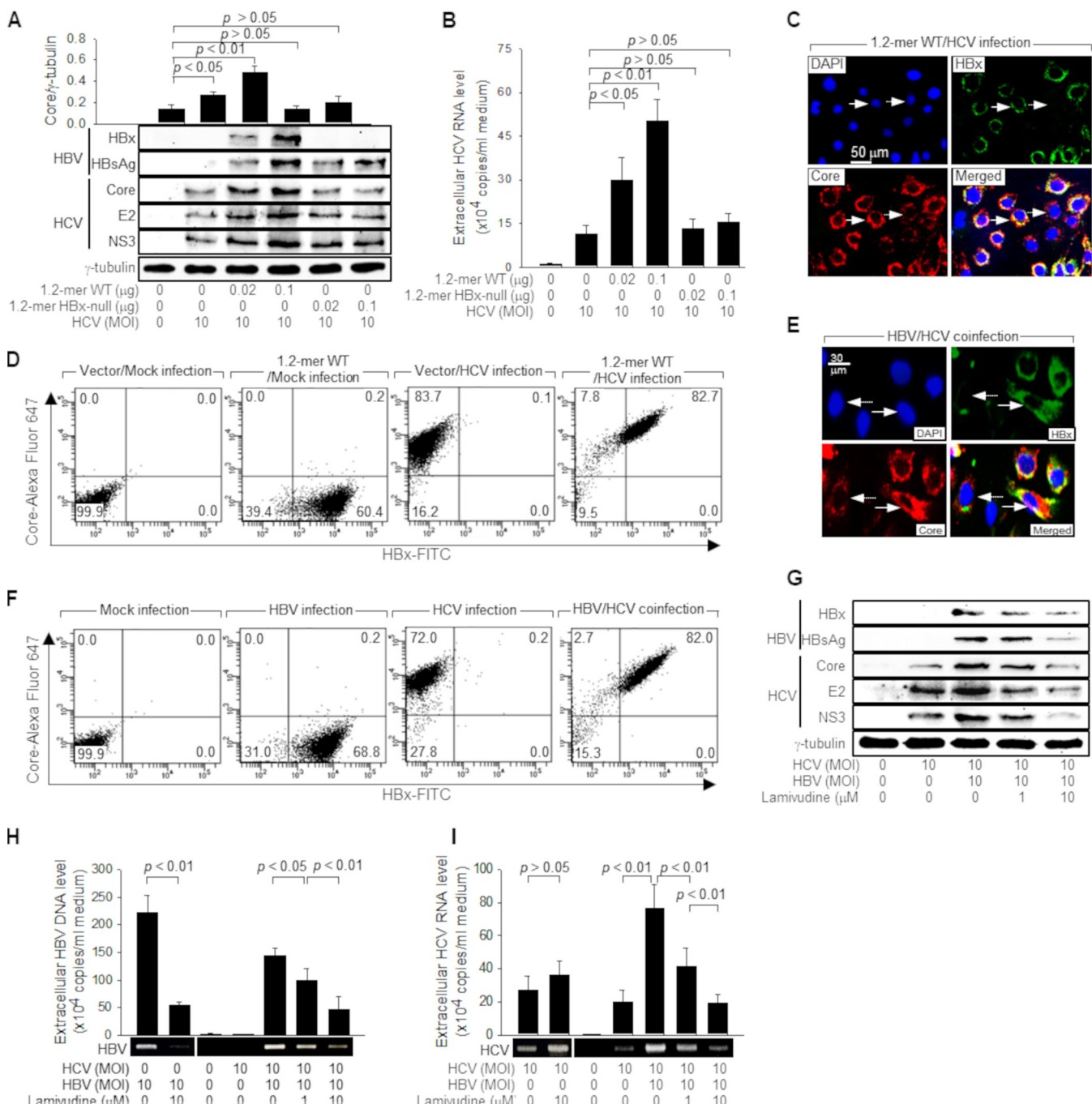

**FIG 1** HBV stimulates HCV replication during coinfection in human hepatoma cells. (A) Huh7D cells were transfected with the indicated amounts of a 1.2-mer HBV replicon (1.2-mer WT) or its HBx-null counterpart (1.2-mer HBx-null) for 24 h and then infected with HCV at an MOI of 10 for an additional 24 h, and the cells were then collected for Western blot analysis. The protein bands were quantified using ImageJ image analysis software (NIH) to indicate the level of HCV core protein, which was normalized to the loading control ($\gamma$-tubulin). Data represent the means ± standard deviations from four independent experiments ($n = 4$). (B) Levels of extracellular HCV RNA from Huh7D cells prepared as described above for panel A were measured by quantitative real-time RT-PCR (qRT-PCR) ($n = 5$). (C) Huh7D cells that were grown on coverslips were transfected with the 1.2-mer WT replicon, infected with HCV as described above for panel A, fixed, and processed for double-label indirect immunofluorescence. Cells were incubated with anti-HBx monoclonal and anti-HCV core protein polyclonal antibodies and then reacted with anti-mouse IgG–FITC and anti-rabbit IgG–rhodamine antibodies to visualize HBx (green) and HCV core protein (red), respectively. The arrow indicates one of the coinfected cells expressing both HBx and HCV core protein, whereas the broken arrow indicates an HCV-infected cell expressing HCV core protein. Nuclei (blue) were stained with 4′,6-diamidino-2-phenylindole (DAPI). (D) Huh7D-NTCP cells were transfected with 0.5 $\mu$g of an empty vector or the 1.2-mer WT replicon for 24 h and then either mock infected or infected with HCV at an MOI of 10 for an additional 48 h. Flow cytometry dot plots show the HBx-FITC signal on the $x$ axis and the HCV core protein-Alexa Fluor 647 signal on the $y$ axis. Lines demarcate quadrants of negative and positive signals for the two dyes, and the numbers at each corner indicate the percentage of cells per quadrant. (E) Huh7D-NTCP cells were coinfected with HBV and HCV, each at an MOI of 10, for 24 h, followed by an immunofluorescence assay as described above for panel C. (F) Huh7D-NTCP cells were either mock infected or

60.2% and 41.2%, respectively, as demonstrated by the HBx positivity in these cells (see Fig. S1D at the URL mentioned above). Data from the flow cytometric analysis also showed a relatively high HBV infection rate in Huh7D-NTCP cells, as demonstrated by the 60 to 70% HBx positivity in the monoinfected cells and the >80% HBx positivity in the coinfected cells (Fig. 1D and F; see also Fig. S2A and B at https://dv.pusan.ac.kr/SynapDocViewServer/viewer/doc .html?key=402882e382e741b101845c0770095ac9&convType=html&convLocale=ko& contextPath=/SynapDocViewServer/). In addition, data from the measurement of extracellular HBV particles in the culture supernatant provided evidence for the replication of HBV in Huh7D cells (Fig. 1H; see also Fig. S1B at the URL mentioned above). HCV replication was also evidenced by the detection of intracellular HCV proteins, including core, E2, and NS3 (Fig. 1A); the subcellular localization of HCV core protein in the infected cells (Fig. 1C and E); and the measurement of extracellular HCV particles in the culture supernatant (Fig. 1B). It was possible to obtain up to a 90% HCV infection rate in Huh7D cells by infecting cells at a multiplicity of infection (MOI) of 10 genome equivalents (GEQ) for 48 h, as demonstrated by the flow cytometric analysis of the HCV core protein in the monoinfected and coinfected cells (Fig. 1D and F; see also Fig. S2A and B at the URL mentioned above), which is consistent with the results of a previous report demonstrating that Huh7D cells are highly permissive for HCV replication (32). Interestingly, the levels of intracellular HCV proteins and extracellular HCV particles were upregulated by HBV derived from the 1.2-mer WT replicon (Fig. 1A and B) and *in vitro* infection systems (Fig. 1G and I). In addition, according to data from the flow cytometric analysis, the average fluorescence signal of the HCV core protein from the coinfected cells was higher than that from the cells infected with HCV alone (Fig. 1D and F; see also Fig. S2A and B at the URL mentioned above). These results indicate that HBV stimulates HCV propagation in cells coinfected with both viruses.

Next, we examined whether HBV replication is required for the stimulation of HCV replication in coinfected cells. Lamivudine is a pyrimidine analog with the potential to inhibit HBV polymerase and thus decrease HBV replication in patients with chronic hepatitis B (34). Treatment with lamivudine successfully inhibited HBV replication in coinfected cells, as demonstrated by the lower levels of intracellular HBV proteins and extracellular HBV particles in the presence of lamivudine (Fig. 1G and H). Lamivudine also strikingly downregulated the levels of intracellular HCV proteins and extracellular HCV particles released from the coinfected cells (Fig. 1G and I), whereas the effect was negligible in cells infected with HCV alone (Fig. 1I). Based on these observations, we conclude that HBV replication is required for the stimulation of HCV replication in coinfected cells.

**HBx upregulates HCV core protein levels to stimulate HCV replication.** We investigated how HBV stimulates HCV replication during coinfection. The effects of HBV on HCV replication were much lower or negligible when an HBx-null HBV replicon, 1.2-mer HBx-null, was used (Fig. 1A and B), suggesting the possible role of HBx in the stimulation of HCV replication. To verify that HBx is responsible for the HBV-mediated stimulation of HCV replication, we investigated whether HBx alone can stimulate HCV replication in Huh7D cells. Indeed, ectopic HBx expression was sufficient to upregulate the levels of intracellular HCV proteins and extracellular HCV particles (Fig. 2A and B). Therefore, we conclude that HBx stimulates HCV replication during HBV/HCV coinfection in Huh7D cells.

Next, we attempted to elucidate the mechanism by which HBx stimulates HCV replication in Huh7D cells. In addition to its role as a major capsid protein, HCV core protein has been strongly implicated in viral replication, where it acts as a positive regulator of viral replication (27, 35). Interestingly, HBx from a 1.2-mer HBV replicon, the infected HBV, or ectopic expression increased the HCV core protein levels in HCV-infected cells (Fig. 1A and G and Fig. 2A). To

**FIG 1** Legend (Continued)
infected with HBV at an MOI of 30 and/or HCV at an MOI of 10 for 48 h. The infected cells were analyzed by flow cytometry as described above for panel D. (G) Huh7D-NTCP cells were coinfected with HBV and HCV, each at an MOI of 10, for 1 h, washed, and incubated for an additional 24 h, followed by Western blotting. For lanes 4 and 5, cells were treated with lamivudine at the indicated concentrations during infection. (H) Levels of extracellular HBV DNA from cells prepared as described above for panel G were measured by either conventional PCR or quantitative real-time PCR (qPCR) ($n = 5$). (I) Extracellular HCV RNA levels from cells prepared as described above for panel G were measured by either conventional RT-PCR or qRT-PCR ($n = 6$).

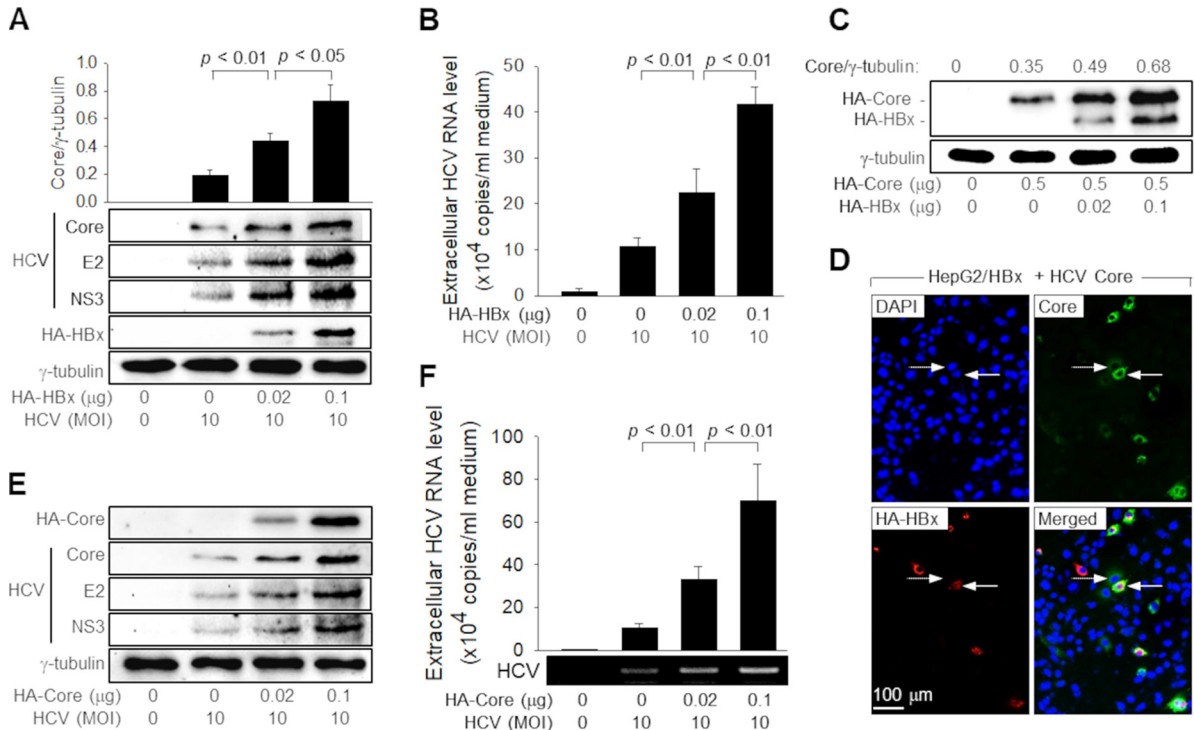

**FIG 2** HBx upregulates HCV core protein levels to stimulate HCV replication. (A) Huh7D cells were transfected with an HBx expression plasmid (HA-HBx) for 24 h and then infected with HCV at an MOI of 10 for an additional 24 h. Subsequently, the cells were collected for Western blotting ($n = 4$). (B) Levels of extracellular HCV RNA from cells prepared as described above for panel A were measured by qRT-PCR ($n = 4$). (C) HepG2 cells were transfected with either an empty vector or an HCV core protein expression plasmid (HA-Core) along with the indicated amounts of HA-HBx. The N-terminal HA tags of HBx and HCV core protein on the same blot were detected using an anti-HA antibody. (D) HepG2 cells were cotransfected with HA-HBx and pCI-neo-core K (encoding untagged HCV core protein) for 48 h and processed for double-label indirect immunofluorescence. Cells were incubated with anti-HA polyclonal and anti-HCV core protein monoclonal antibodies and then reacted with anti-rabbit IgG–rhodamine and anti-mouse IgG–FITC antibodies to visualize HA-HBx (red) and HCV core protein (green), respectively. The arrow indicates one of the cells expressing both HA-HBx and HCV core protein, whereas the broken arrow indicates a cell expressing HCV core protein alone. (E) Huh7D cells were transfected with the indicated amounts of HA-Core for 24 h and then infected with HCV at an MOI of 10 for an additional 24 h. (F) Extracellular HCV RNA levels from cells prepared as described above for panel E were measured as described in the legend of Fig. 1B ($n = 3$).

exclude the possibility that the increase in the HCV core protein levels simply reflects the HBx-mediated stimulation of HCV replication, we examined whether HBx directly upregulated HCV core protein levels in human hepatoma cells. Indeed, HBx upregulated HCV core protein levels in the absence of other HCV proteins in HepG2 cells (Fig. 2C). Immunofluorescence data also clearly showed that HBx strengthened the HCV core protein signal in the cytoplasm of HepG2 cells (Fig. 2D). In addition, the HBx protein derived from either a 1.2-mer HBV replicon or HBV strengthened the HCV core protein signal in the cytoplasm of the HCV-infected cells (Fig. 1C and E).

Having established that HBx upregulates the HCV core protein levels in human hepatoma cells, we attempted to overexpress a hemagglutinin (HA)-tagged HCV core protein (HA-Core) in HCV-infected cells to show that an increase in HCV core protein levels is sufficient for the stimulation of HCV propagation. Indeed, the ectopic expression of HCV core protein dose-dependently upregulated the intracellular levels of HCV proteins, including the HCV core, E2, and NS3 proteins, in the Huh7D cells that were infected with HCV (Fig. 2E). In addition, the ectopic expression of HA-Core upregulated the levels of extracellular HCV particles released from the infected cells up to approximately 7-fold (Fig. 2F). Based on these observations, we conclude that HBx stimulates HCV replication by upregulating HCV core protein levels in human hepatoma cells.

**HBx protects HCV core protein from ubiquitin-dependent proteasomal degradation by downregulating E6AP levels.** Next, we attempted to elucidate the mechanism by which HBx upregulates HCV core protein levels. Previous reports have demonstrated that HCV core protein levels are largely regulated at the posttranslational level through ubiquitin-dependent proteasomal degradation (23, 24, 36). Therefore, we first examined whether

HBx affects the stability of the HCV core protein by comparing the rates of turnover of the HCV core protein in the presence and absence of HBx after treatment of the cells with cycloheximide (CHX), a protein translation inhibitor that interacts with the translocase enzyme to block protein synthesis in eukaryotes (37). The HCV core protein from ectopic expression in HepG2 cells was relatively unstable (half-life [$t_{1/2}$] = 73.5 min), with two-thirds of it being degraded within 2 h after CHX treatment (Fig. 3A, lanes 1 to 4). However, it became more stable in the presence of HBx ($t_{1/2}$ = 259.9 min), leaving approximately 70% of the protein intact under the same conditions (Fig. 3A, lanes 5 to 8). The stability of HCV core protein derived from HCV replication was also higher in the presence of HBx from a 1.2-mer HBV replicon in Huh7D cells or the infected HBV in Huh7D-NTCP cells (Fig. 3B and C), indicating that HBx stabilizes HCV core protein during HBV/HCV coinfection.

Having established that HBx stabilizes HCV core protein, we examined whether HBx affects the ubiquitination of HCV core protein in HepG2 cells. Two strong bands of approximately 120 kDa and 150 kDa and several weaker bands, most likely representing the polyubiquitinated forms of the HCV core protein, were detected in HepG2 cells expressing the HCV core protein (Fig. 3D and E, lanes 2). HBx drastically weakened these bands, resulting in the accumulation of intact HCV core protein (Fig. 3D and E, lanes 3). The polyubiquitinated forms of the HCV core protein were also detected in Huh7D and Huh7D-NTCP cells infected with HCV alone (Fig. 3F and G, lanes 2), whereas these bands became weakened in the presence of HBx derived from either a 1.2-mer HBV replicon or the infected HBV (Fig. 3F and G, lanes 3). These results indicate that HBx upregulates HCV core protein levels by inhibiting its ubiquitin-dependent proteasomal degradation during HBV/HCV coinfection.

Previous reports have demonstrated that E6AP, as an E3 ligase, induces the ubiquitination of HCV core protein (23, 27). The transient expression of the HCV core protein downregulated the E6AP levels in HepG2 cells (Fig. 3H and I, lanes 2), which is consistent with the results of a previous report demonstrating that the HCV core protein inhibits *E6AP* expression to escape ubiquitin-dependent proteasomal degradation (27). Monoinfection with HCV also downregulated E6AP levels in Huh7D and Huh7D-NTCP cells, presumably due to the action of HCV core protein (Fig. 3J and K, lanes 2). In addition, HBx from ectopic expression, a 1.2-mer HBV replicon, or the infected HBV downregulated E6AP levels in HepG2 (Fig. 3H, lane 3), Huh7D (Fig. 3J, lane 3), and Huh7D-NTCP (Fig. 3K, lane 3) cells, respectively. The E6AP levels were further downregulated when HBx and HCV core protein were expressed together by transient transfection in HepG2 cells (Fig. 3I, lane 3) and HBV/HCV coinfection established by either the transient transfection of a 1.2-mer HBV replicon in Huh7D cells (Fig. 3J, lane 4) or infection with HBV in Huh7D-NTCP cells (Fig. 3K, lane 4), resulting in a dramatic decrease in the ubiquitination of the HCV core protein and the subsequent upregulation of the intact HCV core protein in the presence of HBx (Fig. 3D to G, lanes 3). Moreover, ectopic *E6AP* expression restored the ubiquitination of the HCV core protein in the presence of HBx (Fig. 3D, F, and G, lanes 4) and downregulated the levels of the intact HCV core protein in both coexpression (Fig. 3I, lanes 4 and 5) and coinfection (Fig. 3J and K, lanes 5) systems. In addition, E6AP knockdown decreased the ubiquitination of the HCV core protein and upregulated the levels of the intact HCV core protein in the absence of HBx, whereas neither was observed in the presence of HBx (Fig. 3E, lanes 4 and 5). Consistent with these observations, treatment with a proteasome inhibitor, MG132, almost completely abolished the potential for HBx to upregulate HCV core protein levels in both coexpression and coinfection systems (Fig. 3L and M). Interestingly, the ectopic expression of *E6AP* also downregulated HBx levels in both transient-expression (Fig. 3I, lanes 4 and 5) and coinfection (Fig. 3J and K, lanes 5) systems, whereas E6AP knockdown upregulated HBx levels in cells coinfected with HBV and HCV (Fig. 3E, lanes 3 and 5), suggesting that E6AP also acts as an E3 ligase of HBx. Therefore, we conclude that HBx upregulates the levels of the HCV core protein by protecting it from ubiquitin-dependent proteasomal degradation via the downregulation of E6AP levels.

**HBx inhibits *E6AP* expression via DNA methylation to upregulate HCV core protein levels.** We investigated the mechanism by which HBx downregulates E6AP levels in the presence of HCV core protein. Consistent with the results of a previous report (27), HCV core protein upregulated the protein levels and enzyme activities of DNMT1 and -3b and induced the promoter hypermethylation of the *E6AP* gene, resulting in the downregulation of

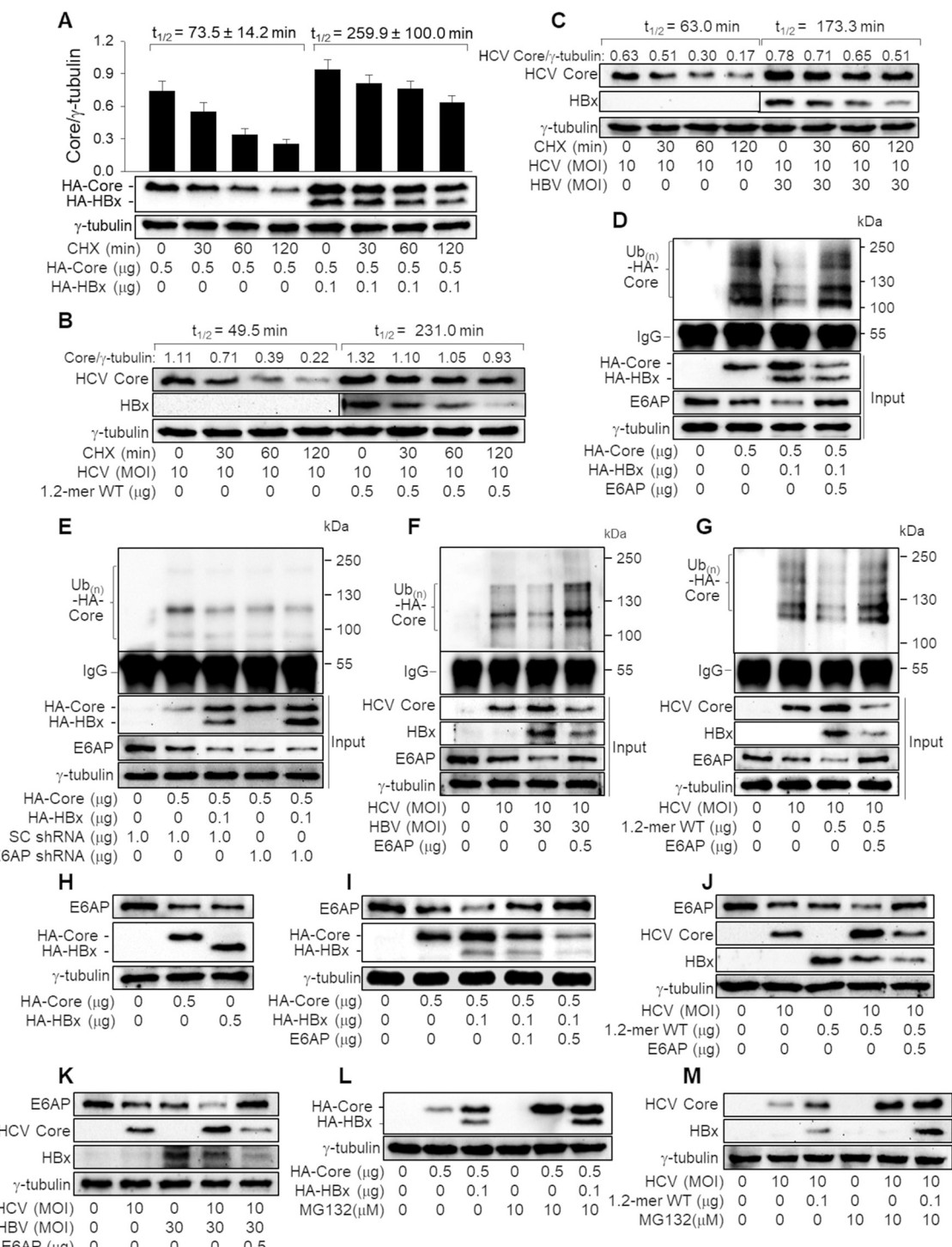

**FIG 3** HBx stabilizes HCV core protein by protecting it from ubiquitin-dependent proteasomal degradation. (A) HepG2 cells were transfected with HA-Core with or without HA-HBx for 48 h, treated with 50 $\mu$M cycloheximide (CHX) for the indicated times, and collected for Western blotting. The band intensity of HA-Core was quantified as described in the legend of Fig. 1A to determine the half-life ($t_{1/2}$) of the HCV core protein ($n = 4$). (B) Huh7D cells were transfected with the 1.2-mer WT replicon and then infected with HCV at an MOI of 10 for an additional 48 h. The $t_{1/2}$ value of the HCV core protein was determined as described above for panel A. (C) Huh7D-NTCP cells were coinfected with HBV at an MOI of 30 and HCV at an MOI of 10 for 48 h. The $t_{1/2}$ value of the HCV core protein was determined as described above for panel A. (D) HepG2 cells were transfected with HA-HBx, HA-Core, an E6AP expression plasmid, and HA-ubiquitin (HA-Ub) for 48 h. Total HA-Core proteins were immunoprecipitated with an anti-HCV core protein antibody and subjected to Western blotting using an anti-HA antibody to detect polyubiquitinated HA-Core. The input shows the levels of the indicated proteins in whole-cell lysates. (E) HepG2 cells were transfected with the indicated amounts of HA-HBx, HA-Core, scrambled (SC) shRNA, and E6AP shRNA for 48 h, and samples were collected for immunoprecipitation (IP), as described above for panel D. (F) Huh7D-NTCP cells were transfected with an empty vector, an E6AP expression plasmid, and

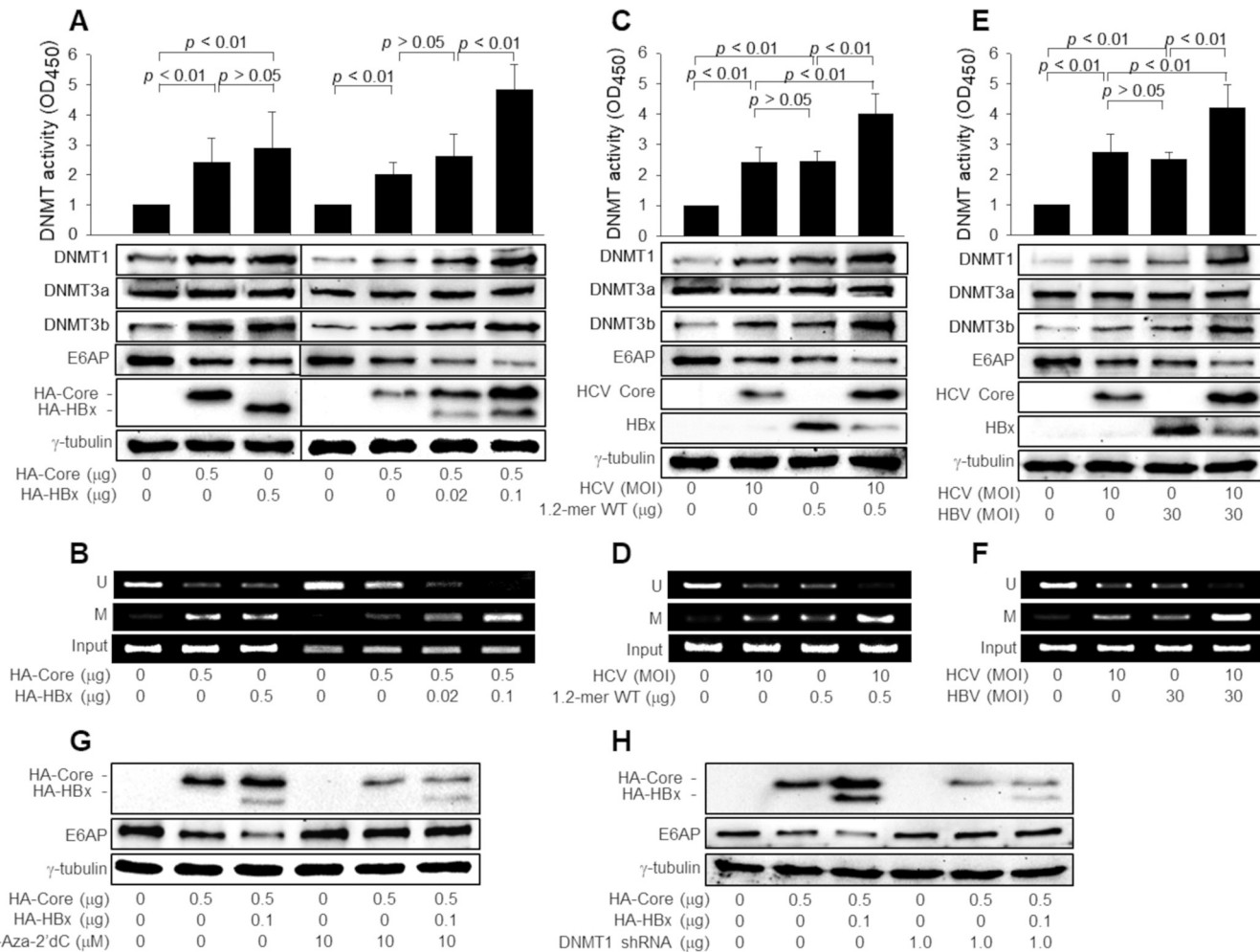

**FIG 4** HBx upregulates HCV core protein levels by repressing E6AP expression via DNA methylation. (A) HepG2 cells were transfected with the indicated amounts of HA-Core and HA-HBx for 48 h. DNA methyltransferase (DNMT) activity in the cell extracts was determined ($n = 6$). The levels of the indicated proteins were determined by Western blotting. (B) Methylation-specific PCR (MSP) was performed to determine whether the CpG sites within the *E6AP* promoter were unmethylated (U) or methylated (M) in cells prepared as described above for panel A. (C) Huh7D cells were transfected with the 1.2-mer WT replicon for 24 h and then infected with HCV at an MOI of 10 for an additional 48 h. DNMT activity in the cell extracts was determined ($n = 4$). The levels of the indicated proteins were determined by Western blotting. (D) MSP was performed as described above for panel B with cells prepared as described above for panel C. (E) Huh7D-NTCP cells were either mock infected or infected with HBV at an MOI of 30 and/or HCV at an MOI of 10 for 48 h. The DNMT activity in the cell extracts was determined ($n = 4$). (F) MSP was performed as described above for panel B with cells prepared as described above for panel E. (G) HepG2 cells were transfected with the indicated amounts of HA-Core and HA-HBx for 48 h. For lanes 4 to 6, cells were treated with 10 $\mu$M 5-aza-2'dC for 24 h before harvest. (H) HepG2 cells were transfected with the indicated amounts of HA-Core and HA-HBx in the presence and absence of the DNMT1 shRNA plasmid for 48 h. OD$_{450}$, optical density at 450 nm.

E6AP levels in both core overexpression (Fig. 4A and B, lanes 2) and HCV infection (Fig. 4C to F, lanes 2) systems. HBx also activated cellular DNA methylation systems to induce the promoter hypermethylation of the *E6AP* gene and the subsequent downregulation of E6AP levels in HBx overexpression (Fig. 4A and B, lanes 3), 1.2-mer HBV replicon (Fig. 4C and D,

**FIG 3** Legend (Continued)
HA-Ub for 24 h and then monoinfected with HCV or coinfected with HBV and HCV for 48 h as described above for panel C, and samples were collected for IP as described above for panel D. (G) Huh7D cells were transfected with an empty vector, the 1.2-mer WT replicon, an E6AP expression plasmid, and HA-Ub for 24 h and then infected with HCV at an MOI of 10 for an additional 48 h. (H) HepG2 cells were transfected with either HA-HBx or HA-Core, followed by Western blot analysis. (I) HepG2 cells were transfected with the indicated amounts of HA-HBx, HA-Core, and an E6AP expression plasmid for 48 h. (J) Huh7D cells were transfected with the 1.2-mer WT replicon and an E6AP expression plasmid for 24 h and then infected with HCV at an MOI of 10 for an additional 48 h. (K) Huh7D-NTCP cells were transfected with either an empty vector or an E6AP expression plasmid for 24 h and then either mock infected or infected with HBV at an MOI of 30 and/or HCV at an MOI of 10 for an additional 48 h. (L) HepG2 cells were transfected with the indicated amounts of HA-HBx and HA-Core for 44 h and then either mock treated or treated with 10 $\mu$M MG132 for an additional 4 h. (M) HepG2 cells were transfected with the 1.2-mer WT replicon and then infected with HCV as described above for panel J. For lanes 4 to 6, cells were either mock treated or treated with 10 $\mu$M MG132 for 4 h before harvest.

lanes 3), and HBV infection (Fig. 4E and F, lanes 3) systems. HBx augmented the potential for the HCV core protein to activate DNMT1 and -3b, induce the promoter hypermethylation of the *E6AP* gene, and downregulate E6AP levels in both coexpression and coinfection systems (Fig. 4A to F). In addition, either treatment with a universal DNMT inhibitor, 5-aza-2′-deoxycytidine (5-aza-2′dC), or knockdown of DNMT1 upregulated the E6AP levels in the presence of the HCV core protein (Fig. 4G and H, lanes 5). This effect was more dramatic in the presence of both HBx and HCV core protein (Fig. 4G and H, lanes 6), whereas it was negligible in the absence of HBx and HCV core protein (Fig. 4G and H, lanes 4), in which the *E6AP* promoter was largely unmethylated (Fig. 4B, lane 1). Accordingly, the inhibition of the cellular DNA methylation system by either 5-aza-2′dC treatment or DNMT1 knockdown downregulated HCV core protein levels in the presence and absence of HBx, abolishing the potential for HBx to upregulate HCV core protein levels (Fig. 4G and H, lanes 5 and 6). Based on these observations, we conclude that HBx protects HCV core protein from ubiquitin-dependent proteasomal degradation by inhibiting *E6AP* expression via DNA methylation.

**HBx stimulates HCV replication by inhibiting *E6AP* expression via DNA methylation during HBV/HCV coinfection.** We investigated whether the repression of *E6AP* expression via DNA methylation was responsible for the HBx-mediated stimulation of HCV replication during coinfection. The restoration of E6AP levels by ectopic *E6AP* expression, 5-aza-2′dC treatment, or DNMT1 knockdown downregulated the HCV core protein levels during HCV infection, resulting in the inhibition of HCV replication, as evidenced by the decreases in the levels of intracellular E2 and NS3 proteins and extracellular HCV particles (Fig. 5A to L, lanes 4). The restoration of E6AP levels also abolished the potential for HBx to upregulate HCV core protein levels and stimulate HCV replication in both HBx overexpression (Fig. 5A to F, lanes 5) and coinfection (Fig. 5G to L, lanes 5) systems. In addition, E6AP knockdown upregulated HCV core protein levels during HCV infection, resulting in the stimulation of HCV replication, as evidenced by the increases in the levels of intracellular E2 and NS3 proteins and extracellular HCV particles (Fig. 5M to P, lanes 4). E6AP knockdown also abolished the potential for HBx to upregulate HCV core protein levels and stimulate HCV replication in both HBx overexpression (Fig. 5M and N, lanes 5) and coinfection (Fig. 5O and P, lanes 5) systems. Therefore, we conclude that HBx stimulates HCV replication during HBV/HCV coinfection by inhibiting *E6AP* expression via DNA methylation.

## DISCUSSION

The establishment of an efficient cell culture system seems to be an essential prerequisite for investigations of the interactions between HBV and HCV during coinfection *in vitro*. It has been demonstrated that the cell culture-adapted HCV JFH1 strain replicates efficiently in Huh7.5, Huh7.5.1, and Huh7D cells derived from the human hepatoma cell line Huh7 (32, 38, 39), producing nearly $10^5$ infectious units per mL within 48 h (39). Consistently, it was possible to detect similar levels of HCV virions in the culture supernatant of Huh7D cells infected with HCV at an MOI of 10 for 24 h (Fig. 1B). The discovery of NTCP as the HBV receptor made it possible to create hepatoma cell lines susceptible to HBV replication (33). However, the cell culture systems for HBV infection often require extremely high viral titers (1,000 to 10,000 GEQ per cell) and fairly long incubation times (over 7 days) to achieve acceptable infection (40, 41). Recently, optimizing infection conditions, for example, by including dimethyl sulfoxide (DMSO) and polyethylene glycol 8000 (PEG 8000) in culture media, made it possible to induce productive HBV infection with low starting inoculum concentrations (10 to 100 GEQ per cell) and short incubation periods (4 to 7 days) (42–44). Consistently, it was possible to detect minimum levels of intracellular HBsAg and extracellular virions from Huh7D-NTCP cells infected with HBV at an MOI of 30 for 24 h (see Fig. S3A and B at https://dv.pusan.ac.kr/SynapDocViewServer/viewer/doc.html?key=402882e382e741b101845c07a1aa5aca&convType=html&convLocale=ko&contextPath=/SynapDocViewServer/). The IFA also detected a small fraction of cells (about 1%) as HBx positive under this condition (data not shown), indicating that an MOI of 30 GEQ corresponds to an MOI of approximately 0.01 in terms of infectious units. Prolonged incubation steadily increased the levels of both intracellular HBsAg and extracellular virions (see Fig. S3A and B at the URL mentioned above). Accordingly, several indicators of productive HBV infection, including intracellular proteins such as HBx and

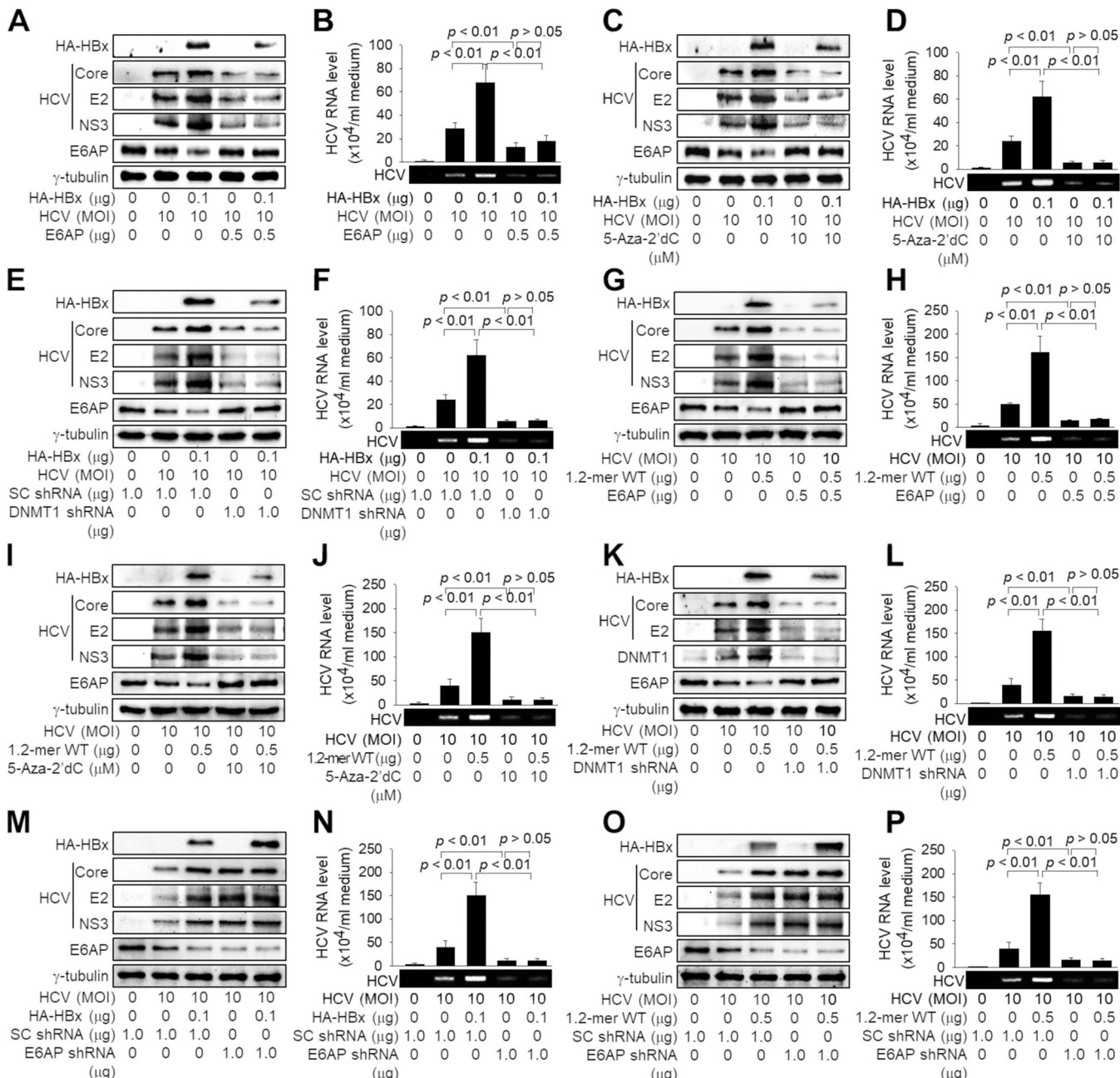

**FIG 5** HBx stimulates HCV replication by repressing E6AP expression via DNA methylation. (A) Huh7D cells were transfected with HA-HBx and an E6AP expression plasmid for 24 h and then infected with HCV at an MOI of 10 for an additional 48 h, followed by Western blotting. (B) Levels of extracellular HCV particles from Huh7D cells prepared as described above for panel A were measured by qRT-PCR as described in the legend of Fig. 1B (*n* = 4). (C) Huh7D cells were transfected with HA-HBx and then infected with HCV as described above for panel A, in the presence or absence of 10 μM 5-aza-2'dC. (D) Levels of extracellular HCV RNA from Huh7D cells prepared as described above for panel C were measured as described in the legend of Fig. 1B (*n* = 5). (E) Huh7D cells were transfected with HA-HBx and a DNMT1 shRNA plasmid and then infected with HCV as described above for panel A, followed by Western blotting. (F) Levels of extracellular HCV RNA from Huh7D cells prepared as described above for panel E were measured as described in the legend of Fig. 1B (*n* = 5). (G) Huh7D cells were transfected with the 1.2-mer WT replicon and an E6AP expression plasmid and then infected with HCV as described above for panel A. (H) Levels of extracellular HCV RNA from cells prepared as described above for panel G were measured as described in the legend of Fig. 1B (*n* = 3). (I) Huh7D cells were transfected with the 1.2-mer WT replicon and then infected with HCV as described above for panel A. For lanes 4 and 5, cells were treated with 10 μM 5-aza-2'dC, as described in the legend of Fig. 4G. (J) Levels of extracellular HCV RNA from cells prepared as described above for panel I were measured as described in the legend of Fig. 1B (*n* = 5). (K) Huh7D cells were transfected with the 1.2-mer WT replicon and a DNMT shRNA plasmid and then infected with HCV as described above for panel A. (L) Levels of extracellular HCV RNA from cells prepared as described above for panel K were measured as described in the legend of Fig. 1B (*n* = 5). (M) Huh7D cells were transfected with HA-HBx and an E6AP shRNA plasmid and then infected with HCV as described above for panel A, followed by Western blotting. (N) Levels of extracellular HCV RNA from Huh7D cells prepared as described above for panel M were measured as described in the legend of Fig. 1B (*n* = 5). (O) Huh7D cells were transfected with the 1.2-mer WT replicon and E6AP shRNA and then infected with HCV as described above for panel A. (P) Levels of extracellular HCV RNA from cells prepared as described above for panel O were measured as described in the legend of Fig. 1B (*n* = 5).

HBsAg (see Fig. S1A at https://dv.pusan.ac.kr/SynapDocViewServer/viewer/doc.html?key=402882e382e741b101845c07289e5ac8&convType=html&convLocale=ko&contextPath=/SynapDocViewServer/), extracellular virions in the culture supernatant (see Fig. S1B at the URL mentioned above), and cccDNA in Hirt extracts (see Fig. S1C at the URL mentioned above), were obviously detected from Huh7D-NTCP cells infected with HBV at an MOI of 30 for 48 h by Western blotting, quantitative real-time PCR (qPCR), and Southern blotting, respectively. According to data from the IFA, approximately 40% of the cells turned out to be HBx positive in cells infected with HBV at an MOI of 30 for 48 h (see Fig. S1D at the URL mentioned above). Data from the flow cytometry analysis also showed 60 to 70% HBx positivity in cells monoinfected with HBV at an MOI of 30 for 48 h and over 80% HBx positivity in cells coinfected with HBV and HCV at MOIs of 30 and 10, respectively, for 48 h (Fig. 1D and F; see also Fig. S2A and B at https://dv.pusan.ac.kr/SynapDocViewServer/viewer/doc.html?key=402882e382e741b101845c0770095ac9&convType=html&convLocale=ko&contextPath=/SynapDocViewServer/). Although treatment with 4% PEG 8000 dramatically increased HBV replication, possibly by increasing the rate of infection and promoting virus spread (see Fig. S1F at the URL mentioned above), as demonstrated in a previous report (42), it was not absolutely required for HBV replication in Huh7D-NTCP cells. Therefore, it was possible to establish an *in vitro* HBV infection system in the present study by infection with HBV at MOIs of 10 to 30 for 24 to 48 h in the absence of PEG 8000, which was also compatible with HCV infection.

Most clinical and experimental studies have demonstrated that HCV is dominant over HBV during coinfection (5, 6, 10, 12, 14–16, 18–20, 34, 45, 46). Considering the possible roles of HBx and HCV core protein in the replication of HBV and HCV, respectively, the reciprocal regulation between HBx and HCV core protein may contribute to viral interference and the determination of a dominant virus during coinfection. Early studies by Shih et al. demonstrated that the HCV core protein strongly inhibits HBV replication (17, 18). Those researchers further demonstrated that the HCV core protein interacts with HBV polymerase and inhibits its binding to the HBV package signal, preventing the encapsidation of HBV pregenomic RNA (pgRNA) into its capsid (19). It was also hypothesized that HCV core protein directly interacts with HBx to suppress HBV enhancer activity and inhibit HBV transcription (19, 20). More recently, it was demonstrated that the HCV core protein inhibits HBV replication by downregulating HBx levels via Siah-1-mediated proteasomal degradation during coinfection (21). Consistently, the present study showed that HCV core protein downregulates HBx levels to inhibit HBV propagation in Huh7D-NTCP cells (Fig. 1H and Fig. 3K). While related studies so far have focused on the role of HCV core protein in the suppression of HBV replication, the present study attempted to elucidate the role of HBx in the regulation of HCV replication, providing evidence that HBx stimulates HCV replication by upregulating the levels of HCV core protein (Fig. 2). The role of HBx as a positive regulator of HCV replication was further assessed by treatment with lamivudine, a pyrimidine analog that inhibits HBV DNA reverse transcriptase (34), which inhibited HCV replication in the presence but not in the absence of HBV (Fig. 1I). For this effect, HBx inhibited *E6AP* expression via DNA methylation and inhibited the ubiquitin-dependent proteasomal degradation of HCV core protein (Fig. 3 and 4), providing more nucleocapsid proteins to facilitate capsid assembly and accelerate HCV replication. The potential for HBx to downregulate HCV core protein levels was not detectable when the difference in E6AP levels was removed by ectopic expression, 5-aza-2′dC treatment, DNMT1 knockdown, or E6AP knockdown (Fig. 3 and 5), suggesting that E6AP is a unique factor that is involved in the HBx-mediated upregulation of HCV core protein levels during coinfection. Thus, the present study provides a new mechanism for HCV dominance during HBV/HCV coinfection. It is uncertain why HBx inhibits *E6AP* expression and stabilizes HCV core protein during HBV/HCV coinfection. Interestingly, the restoration of E6AP levels by ectopic expression, 5-aza-2′dC treatment, or DNMT1 knockdown downregulated the HBx levels, whereas E6AP knockdown upregulated the HBx levels (Fig. 3E and Fig. 5I and L), suggesting that E6AP also targets HBx in addition to HCV core protein to

induce its ubiquitin-dependent proteasomal degradation. According to our preliminary data, E6AP acts as an E3 ligase of HBx to induce its ubiquitin-dependent proteasomal degradation (data not shown). Thus, HBx, as in the case of HCV core protein (27), may downregulate E6AP levels to counteract the host antiviral defense system. It remains uncertain why the downregulation of E6AP levels by HBx and HCV core protein provides more benefits to HCV core protein than to HBx and results in HCV dominance. The proteasomal degradation of HBx appears to be regulated primarily by another E3 ligase, termed Siah-1 (47). As Siah-1 is a target of p53, HBx activates Siah-1 expression via the upregulation of p53 levels, resulting in the downregulation of HBx levels (47). It has also been demonstrated that HCV core protein inhibits HBV replication via the activation of Siah-1 expression during coinfection (21), suggesting that Siah-1 is dominant over E6AP in the regulation of HBx levels. It is thus interesting to investigate the relative roles of Siah-1 and E6AP in the regulation of HBx, which can affect viral dominance during coinfection.

Previous studies on HBV/HCV coinfection have not uniformly reported HCV dominance. Some clinical studies have suggested reciprocal interference or even a dominant role of HBV (48, 49). For example, Zarski et al. found that HCV RNA levels are significantly decreased in HBV/HCV-coinfected patients compared to HBV-negative cases (48). However, some *in vitro* studies have demonstrated that both HBV and HCV replicate together within the same hepatocytes without interference (50, 51), suggesting that any interactions seen clinically are more likely related to host immune responses (50). Moreover, Eyre et al. found that HBV replication is slightly enhanced by HCV core protein, as indicated by the increased HBV DNA release in coinfected cells (51). Therefore, patients with HBV/HCV coinfection may have a large spectrum of virological profiles, although the reason for this is uncertain. One possibility is that HBV and HCV can alternate their dominance during different periods of infection. Indeed, a multicenter longitudinal follow-up study in Italy showed that the patterns of virological responses in these cases are widely divergent and change over time (52). In addition, Eyre et al. suggested that the different outcomes of coinfection are largely attributable to host-derived factors and/or the host immune response, which was based on the observation that the HCV-enhanced expression of interferon in liver cells plays an inhibitory role in HBV replication. It is also possible that viral interactions can vary depending on the combination of HBV and HCV genotypes. For example, a study by Schüttler et al. revealed a 3- to 11-fold inhibition of HBV enhancer regions by HCV core protein derived from multiple viral genotypes (20). Consistent with our preliminary data, some HBx variants did not inhibit *E6AP* expression and were unable to stimulate HCV replication during coinfection (data not shown). More intensive studies on both clinical and experimental aspects of coinfection are required to better understand the interactions between HBV and HCV during coinfection.

## MATERIALS AND METHODS

**Plasmids.** The plasmids pCMV-3×HA1-Core (HA-Core) (53) and pCMV-3×HA1-HBx (HA-HBx) (54) encode the full-length HCV core protein (genotype 1b) and HBx (genotype D), respectively, downstream of the three copies of influenza virus hemagglutinin (HA). The plasmid pCI-neo-core K, which encodes the core region of the HCV-K isolate (genotype 1b) under the control of the human cytomegalovirus immediate early promoter, was described previously (55). The construction of a DNMT1 short hairpin RNA (shRNA) plasmid using the SirenCircle RNA interference (RNAi) system (Allele Biotechnology) was described previously (56). The 1.2-mer WT HBV replicon containing 1.2 U of the HBV genome (genotype D) and its HBx-null counterpart (57) were kindly provided by W.-S. Ryu (Yonsei University, Republic of Korea). The *E6AP* expression plasmid pCMVT N-HA-hE6AP (catalog no. 37601), encoding full-length human HA-tagged E6AP, was purchased from Addgene. The plasmids RC210241 (catalog no. 003049), encoding human NTCP, and pCH110 (catalog no. 27-4508-01), carrying the *Escherichia coli* $\beta$-galactosidase ($\beta$-Gal) gene, were obtained from OriGene and Amersham, respectively. The plasmid pHA-Ub was a gift from Y. Xiong (University of North Carolina at Chapel Hill, Chapel Hill, NC, USA).

**Cell culture and transfection.** The HepG2 cell line (catalog no. 88065), a human hepatoblastoma cell line, was purchased from the Korean Cell Line Bank (KCLB). The Huh7D cell line was kindly provided by S. M. Feinstone (U.S. Food and Drug Administration). A stable cell line, Huh7D-NTCP, was established by transfection with RC210241, followed by selection with 500 $\mu$g · mL$^{-1}$ G418 sulfate (catalog no. A1720; Sigma-Aldrich). For transient expression, $2 \times 10^5$ cells per 60-mm dish were transfected with 2 $\mu$g of the appropriate plasmid(s) using TurboFect transfection reagent (catalog no. R0532; Thermo Fisher Scientific) according to the manufacturer's instructions. All cells were cultured in Dulbecco's modified Eagle

medium (DMEM) (catalog no. LM001-05; WelGENE) supplemented with 10% fetal bovine serum (FBS) (catalog no. FBS-22A; Capricorn), 100 U · mL$^{-1}$ penicillin G (catalog no. P3032-25MU; Sigma-Aldrich), and 100 $\mu$g · mL$^{-1}$ streptomycin (catalog no. 21865; USB). The cells were treated with 5-aza-2′dC (catalog no. A3656; Sigma-Aldrich), cycloheximide (catalog no. C7698; Sigma-Aldrich), lamivudine (catalog no. L1295; Sigma-Aldrich), or MG132 (catalog no. 474790; Millipore) under the conditions indicated in the figure legends.

**Preparation of viral stocks.** For HCV stocks, the plasmid pJFH-1, which contains HCV cDNA (from a Japanese patient with fulminant hepatitis) behind a T7 promoter (58), was linearized at the 3′ end of the HCV cDNA by XbaI digestion. The linearized DNA was then used as a template for *in vitro* transcription (MEGAscript, catalog no. AM1333; Ambion). Huh7D cells were transfected with 10 $\mu$g of JFH1 RNA by electroporation and incubated for 48 h to obtain viral seeds. HCV stocks were prepared by infecting Huh7D cells with HCV seeds at an MOI of 0.01 for 9 days. Hep3B-NTCP cells were transiently or stably transfected with a 1.2-mer HBV replicon plasmid, as described above. For HBV stock preparation, HBV particles in the culture supernatant were further amplified in Hep3B-NTCP cells by infection at an MOI of 10 for 72 h. A virus concentration procedure with PEG 8000 was not adopted to prepare the virus inoculum.

**Viral infection.** Huh7D cells were used for monoinfection by HCV, while Huh7D-NTCP cells were used for either monoinfection by HBV or coinfection by HBV and HCV, as previously described (59). Briefly, $2 \times 10^5$ cells seeded into 60-mm dishes were washed twice with serum-free DMEM, either mock infected or infected with HBV and/or HCV at the indicated MOIs for 24 h, washed twice with serum-free DMEM, and then incubated for an additional 48 h, unless otherwise stated, in DMEM containing 3% FBS and 2% DMSO (catalog no. D8418; Sigma).

**Determination of HBV and HCV levels.** The HBV titers were determined as described previously (60). Briefly, HBV genomic DNA was purified from the precipitated HBV particle-antibody complexes using the QIAamp DNA minikit (catalog no. 51306; Qiagen). For conventional PCR analysis of HBV DNA, the genomic DNA was amplified using 2$\times$ *Taq* PCR master mix 1 (catalog no. ST301-19h; BioFACT) and a primer pair, HBV 1399F (5′-TGG TAC CTC CGC GGG ACG TCC TT-3′) and HBV 1632R (5′-AGC TAG CGT TCA CGG TGT CTC C-3′). For quantitative real-time PCR (qPCR) of HBV, HBV DNA was amplified using SYBR premix Ex *Taq* II (catalog no. RR82LR; TaKaRa Bio) and a primer pair, HBV 379F (5′-GTG TCT GCG GCG TTT TAT CA-3′) and HBV 476R (5′-GAC AAA CGG GCA ACA TAC CTT-3′), in a Rotor-Gene Q PCR machine (Qiagen). For conventional reverse transcription-PCR (RT-PCR), HCV RNA extracted from the culture supernatant using the QIAamp viral RNA minikit (catalog no. 52904; Qiagen) was reverse transcribed with a reverse primer, HCV 290R (5′-AGT ACC ACA AGG CCT TTC G-3′), using the AccuPower RT PreMix kit (catalog no. K-2041; Bioneer). HCV cDNA was then amplified with primers HCV 130S (5′-CGG GAG AGC CAT AGT GG-3′) and HCV 290R, using *Taq* polymerase (catalog no. ST-301-10h; BioFACT) to detect HCV core RNA levels, as previously described (53). Quantitative real-time RT-PCR (qRT-PCR) of HCV RNA levels was performed using primers HCV 130S and HCV 290R and SYBR green PCR master mix (catalog no. RR82LR; TaKaRa), using the Rotor-Gene Q machine (Qiagen), as previously described (59).

**Western blot analysis.** Cells were lysed in buffer (50 mM Tris-HCl [pH 7.5], 150 mM NaCl, 0.1% SDS, 1% NP-40) supplemented with protease inhibitors. The protein concentration of the cell extracts was measured using a protein assay kit (catalog no. 5000006; Bio-Rad). Cell extracts were separated by SDS-PAGE and transferred onto a nitrocellulose membrane (catalog no. 10600003; Amersham). Membranes were then incubated with antibodies against E6AP (catalog no. PA3-843; Thermo Scientific) (1:2,000 dilution), DNMT1 (catalog no. ab19905; Abcam) (1:500 dilution), HCV core proteins (catalog no. ab58713; Abcam) (1:500 dilution), DNMT3a (catalog no. sc-373905; Santa Cruz Biotechnology) (1:500 dilution), DNMT3b (catalog no. sc-81252; Santa Cruz Biotechnology) (1:500 dilution), HA (catalog no. sc-7392; Santa Cruz Biotechnology) (1:500 dilution), HBsAg (catalog no. sc-53300; Santa Cruz Biotechnology) (1:400 dilution), HCV E2 (catalog no. sc-57769; Santa Cruz Biotechnology) (1:500 dilution), HCV NS3 (catalog no. sc-69938; Santa Cruz Biotechnology) (1:500 dilution), ubiquitin (catalog no. sc-9133; Santa Cruz Biotechnology) (1:500 dilution), $\gamma$-tubulin (catalog no. sc-17787; Santa Cruz Biotechnology) (1:500 dilution), and HBx (catalog no. MAB8419; Millipore) (1:500 dilution) and subsequently incubated with a horseradish peroxidase (HRP)-conjugated anti-mouse (catalog no. BR170-6516; Bio-Rad) (1:3,000 dilution), anti-rabbit IgG(H+L) (catalog no. BR170-6515; Bio-Rad) (1:3,000 dilution), or anti-goat IgG(H+L) (catalog no. 31400; Thermo Scientific) (1:10,000 dilution) secondary antibody. The ECL kit (catalog no. K-12043-D20; Advansta) was used to visualize protein bands via the ChemiDoc XRS imaging system (Bio-Rad).

**Immunofluorescence analysis.** Cells that were grown on coverslips under the indicated conditions were fixed in 4% formaldehyde at 20℃ for 15 min and permeabilized in methanol at $-20$℃ for 10 min. The coverslips were then incubated at 20℃ for 3 h with anti-HCV core protein polyclonal (catalog no. ab58713; Abcam) (1:200) and anti-HBx monoclonal (catalog no. sc-57760; Santa Cruz) (1:500) antibodies (Fig. 1C and D; see also Fig. S1D at https://dv.pusan.ac.kr/SynapDocViewServer/viewer/doc.html?key=402882e382e741b101845c07289e5ac8&convType=html&convLocale=ko&contextPath=/SynapDocViewServer/) and with anti-HA polyclonal (catalog no. ab9110; Abcam) (1:200) and anti-HCV core protein monoclonal (catalog no. ab-2740; Abcam) (1:200) antibodies (Fig. 2D). The cells were then incubated with anti-rabbit IgG–rhodamine (catalog no. 31670; Invitrogen) (1:200 dilution) and anti-mouse IgG–fluorescein isothiocyanate (FITC) (catalog no. F0257-1ML; Sigma-Aldrich) (1:100 dilution) antibodies at 20℃ for 1 h. The slides were prepared with UltraCruz mounting medium (catalog no. sc-24941; Santa Cruz Biotechnology) and visualized using an Eclipse fluorescence microscope (Nikon).

**Flow cytometry analysis.** For the determination of intracellular HBx and HCV core protein levels by flow cytometry analysis, cells were fixed using a BD Cytofix/Cytoperm kit (catalog no. 554714; BD Biosciences). Briefly, $1 \times 10^6$ cells were resuspended in a fixation/permeabilization solution at 4℃ for 20 min. A staining buffer containing 0.1% sodium azide and 2% FBS was added to the cell suspension, followed by incubation at 20℃ for 10 min. The cells were reacted with anti-HBx (catalog no. sc-57760; Santa Cruz Biotechnology) (1:20 dilution) and anti-HCV core protein (catalog no. ab58713; Abcam) (1:50 dilution) primary antibodies on ice for 30 min and then with anti-mouse IgG–FITC (catalog no. F0257; Sigma) (1:100 dilution) and anti-rabbit IgG–FSD

674 (catalog no. RSA1265; BioActs) (1:100) secondary antibodies for an additional 30 min in the dark. Cells were washed twice with BD Perm/Wash buffer (BD Biosciences) and then resuspended in a staining buffer or phosphate-buffered saline (PBS). Data were obtained using a FACSAria Fusion sorter (BD Biosciences) with FACSDiva 9.0.1 software (BD Biosciences).

**Southern blot analysis of HBV DNA.** A modified Hirt extraction procedure was used to isolate HBV DNA from the infected cells, as described previously (61). Briefly, cells were incubated in 190 $\mu$L of cell lysis buffer (50 mM Tris-HCl [pH 7.5], 10 mM EDTA, 150 mM NaCl, and 1% SDS) for 30 min at 25°C. The cell lysates were mixed with 190 $\mu$L of 5 M NaCl and incubated at 4°C overnight. Hirt DNA was extracted by phenol-chloroform extraction and dissolved in Tris-EDTA (TE) buffer (10 mM Tris-HCl [pH 8.0], 1 mM EDTA). Southern blotting was performed using the digoxigenin (DIG)-High Prime DNA labeling and detection starter kit (catalog no. 11 745 832 910; Roche), according to the manufacturer's instructions. Briefly, Hirt DNA samples were separated by electrophoresis through an agarose gel and transferred onto a nitrocellulose membrane (catalog no. 10600004; Amersham) by capillary transfer and UV cross-linking. The membrane was hybridized with a DIG-labeled HBV-specific probe, which was prepared by PCR amplification of HBV DNA using primers HBV1399-F (5'-TGG TAC CTG CGC GGG ACG TCC TT-3') and HBV1632-R (5'-AGC TAG CGT TCA CGG TGG TCT CC-3') and DNA labeling with the DIG-High Prime DNA labeling kit (Roche) in hybridization buffer; washed; and visualized using the DIG detection starter kit (Roche).

**Immunoprecipitation.** Immunoprecipitation (IP) assays were performed using a Classic magnetic IP/coimmunoprecipitation (co-IP) assay kit (catalog no. 88804; Thermo Scientific) according to the manufacturer's specifications. Briefly, HepG2 cells (1 × 10$^6$ cells per 100-mm-diameter plate) were transiently transfected with the indicated amounts of eukaryotic expression plasmids encoding HBx, HCV core protein, E6AP, or ubiquitin for 48 h. Whole-cell lysates (900 $\mu$g) were incubated with anti-HCV core protein antibody (catalog no. MAB7949; Abnova) (2 $\mu$g) overnight at 4°C to allow the formation of immune complexes. After intensive washing, the immune complexes were harvested with protein A/G magnetic beads (0.25 mg) by incubation for 1 h with mixing. The beads were then collected using a magnetic stand (Pierce), and the antigen/antibody complexes eluted from them were subjected to Western blotting using an anti-HA antibody.

**DNMT activity assay.** HepG2 cells (2 × 10$^5$ cells per 60-mm-diameter plate) were transiently transfected with the plasmids encoding HBx or HCV core protein for 48 h. DNMT activity in the cell lysates was measured using the EpiQuick DNMT activity/inhibition assay ultrakit (catalog no. P-3009-96; Epigentek) according to the manufacturer's instructions.

**Methylation-specific PCR (MSP).** Genomic DNA was extracted from cells using the QIAamp DNA minikit (catalog no. 51306; Qiagen) according to the manufacturer's instructions. Bisulfite modification of the genomic DNA (1 $\mu$g) was carried out using the EpiTect bisulfite kit (catalog no. 59104; Qiagen). The modified DNA was amplified by PCR using the primer pair E6AP-BS-1F (5'-GGT TAT AGA TAG TAG AAA TT-3') and E6AP-BS-3R (5'-ACC CAA CAC CAC CAT CTT A-3') to obtain the CpG-rich region of the E6AP promoter (positions −275 to +82). The first PCR product was amplified by PCR using the methylated primer pair E6AP-Me-1F (5'-TTT TTA ATG GTT TGT GTG TC-3') and E6AP-Me-1R (5'-TAC AAA CAA CGC ACA CCG-3') and an unmethylated primer pair, E6AP-Un-1F (5'-TTT TTA ATG GTT TGT GTG TT-3') and E6AP-Un-2R (5'-CAC ACA AAT CTC ACA ACC A-3'), as previously described (27).

**Statistical analysis.** Values indicate the means ± standard deviations from at least three independent experiments. Two-tailed Student's $t$ test was used for all statistical analyses. $P$ values of ≤0.05 were considered statistically significant.

## ACKNOWLEDGMENTS

This work was supported by a National Research Foundation of Korea (NRF) grant funded by the South Korean government (MEST) (NRF-2019R1A2C2011478) and the BK21 FOUR Program by a Pusan National University research grant, 2021.

We thank W.-S. Ryu and S. M. Feinstone for providing the HBV replicon system and Huh7D cells, respectively, used in this study.

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
