## [Reviewer comments · Microbiology Spectrum]

Microbiology Spectrum

Hepatitis B virus X protein stimulates hepatitis C virus replication by protecting HCV core protein from E6AP-mediated proteasomal degradation

Hyunyoung Yoon, Jiwoo Han, and Kyung Lib Jang

Corresponding Author(s): Kyung Lib Jang, Pusan National University

Review Timeline:

Submission Date:	May 27, 2022
Editorial Decision:	August 7, 2022
Revision Received:	October 5, 2022
Editorial Decision:	October 8, 2022
Revision Received:	October 11, 2022
Accepted:	October 21, 2022

Editor: Leiliang Zhang

Reviewer(s): Disclosure of reviewer identity is with reference to reviewer comments included in decision letter(s). The following individuals involved in review of your submission have agreed to reveal their identity: Kuanhui Xiang (Reviewer #1)

Transaction Report:

DOI: <https://doi.org/10.1128/spectrum.01432-22>

August 7, 2022

Prof. Kyung Lib Jang
Pusan National University
Microbiology
san 30, Jangjeon-dong, Keumjeong-gu
Busan 46241
Korea (South), Republic of

Re: Spectrum01432-22 (Hepatitis B virus X protein stimulates hepatitis C virus replication by protecting HCV core protein from E6AP-mediated proteasomal degradation)

Dear Prof. Kyung Lib Jang:

Link Not Available

Sincerely,

Leiliang Zhang

Journals Department
Reviewer comments:

Reviewer #1 (Comments for the Author):

Hyunyoung Yoon and Kyung Lib Jang reported in this manuscript that HBV could improve HCV replication through its HBx protein. Mechanically, HBx upregulated both the protein levels and enzyme activities of cellular DNA methyltransferase1(DNMT1) and DNMT3b, and this subsequently reduced the expression levels of the E6-associated protein (E6AP), an E3 ligase of HCV core protein, via DNA methylation. The ubiquitin-dependent proteasomal degradation of the HCV core protein was severely impaired in the presence of HBx, whereas this effect was not observed when either E6AP was ectopically expressed or restored by treatment with 5-Aza-2'dC. Although the authors have made several additional data to

improve the quality of the manuscript, some issues still remain.

Major issues:

1. The authors showed that HBV infection efficacy was very high with low MOI and HBsAg could be detected even after 24 hours infection, which is amazing. The authors also only incubated HBV 1 hour with the cells and lead to that so high infection efficacy. Usually, most studies incubate the viruses for 24 hours.
2. The authors showed in figure 1 that lamivudine treatment for 48 h can reduce HBx protein expression, which is also surprised to me. As we all known, lamivudine inhibits HBV replication via block viral reverse transcription, resulting in low mature viral particles secretion. Why lamivudine treatment could significantly reduce HBx expression only in a 48 h of viral replication period?
3. As the results shown, E6AP could also inhibit HBx expression. how? Does it similar to HCV core protein via the ubiquitin-dependent proteasomal degradation?
4. The authors only showed the HBx could reduce E6AP expression to improve HCV core protein expression. Is there other pathways for HBx stimulate HBV core protein expression besides E6AP? The authors could design the shRNA or siRNA to downregulate E6AP to study if HBx could still influence on HCV replication.
5. As mentioned above, E6AP also inhibit HBx expression. Both HCV core protein and HBx influence on E6AP expression. why only HCV replication was improved rather than HBV? Are there other pathways impact HBV replication?
6. Fig.1C and 1D, why HBx and HCV core protein were shown in different appearance in these two IF images?

Minor issue:

1. Line 87, HCV replication should be "HBV replication".
2. Line 130, why the authors also use GEQ to normalize HCV? Did the author use RT-qPCR to quantify HCV RNA levels to determine HCV amounts of infection? Usually, the MOI of HCV was performed by TCID50.

Reviewer #2 (Comments for the Author):

This study described that HBV X protein stimulated HCV replication by inhibiting the expression of E6-associated protein via DNA methylation. The decreased expression of E6-associated protein protected the HCV core protein from proteasomal degradation, which could contribute to HCV dominance during HBV/HCV coinfection. The experiments have been carried out with care. The reviewer only has 1 major comment that will need to be addressed.

Major comment

1. Since all study works on the coinfection of HBV and HCV, it would be important to show that the stimulation of HCV replication by HBV infection or expression of HBV X protein occurs in the same cells. The reviewer suggests the authors repeat the Figure 1 C and 1D and analyze the expression of HBx and HCV core by flow cytometry.

Minor comment

1. line 87, replace the "inhibits HCV replication" with "inhibits HBV replication".

Staff Comments:

Preparing Revision Guidelines

Please return the manuscript within 60 days; if you cannot complete the modification within this time period, please contact me. If you do not wish to modify the manuscript and prefer to submit it to another journal, please notify me of your decision immediately so that the manuscript may be formally withdrawn from consideration by Microbiology Spectrum.

Hyunyoung Yoon and Kyung Lib Jang reported in this manuscript that HBV could improve HCV replication through its HBx protein. Mechanically, HBx upregulated both the protein levels and enzyme activities of cellular DNA methyltransferase1(DNMT1) and DNMT3b, and this subsequently reduced the expression levels of the E6-associated protein (E6AP), an E3 ligase of HCV core protein, via DNA methylation. The ubiquitin-dependent proteasomal degradation of the HCV core protein was severely impaired in the presence of HBx, whereas this effect was not observed when either *E6AP* was ectopically expressed or restored by treatment with 5-Aza-2'dC. Although the authors have made several additional data to improve the quality of the manuscript, some issues still remain.

Major issues:

1. The authors showed that HBV infection efficacy was very high with low MOI and HBsAg could be detected even after 24 hours infection, which is amazing. The authors also only incubated HBV 1 hour with the cells and lead to that so high infection efficacy. Usually, most studies incubate the viruses for 24 hours.
2. The authors showed in figure 1 that lamivudine treatment for 48 h can reduce HBx protein expression, which is also surprised to me. As we all known, lamivudine inhibits HBV replication via block viral reverse transcription, resulting in low mature viral particles secretion. Why lamivudine treatment could significantly reduce HBx expression only in a 48 h of viral replication period?
3. As the results shown, E6AP could also inhibit HBx expression. how? Does it similar to HCV core protein via the ubiquitin-dependent proteasomal degradation?

4. The authors only showed the HBx could reduce E6AP expression to improve HCV core protein expression. Is there other pathways for HBx stimulate HBV core protein expression besides E6AP? The authors could design the shRNA or siRNA to downregulate E6AP to study if HBx could still influence on HCV replication.
5. As mentioned above, E6AP also inhibit HBx expression. Both HCV core protein and HBx influence on E6AP expression. why only HCV replication was improved rather than HBV? Are there other pathways impact HBV replication?
6. Fig.1C and 1D, why HBx and HCV core protein were shown in different appearance in these two IF images?

Minor issue:

1. Line 87, HCV replication should be "HBV replication".
2. Line 130, why the authors also use GEQ to normalize HCV? Did the author use RT-qPCR to quantify HCV RNA levels to determine HCV amounts of infection?

Usually, the MOI of HCV was performed by TCID50.

Reviewer #1 (Comments for the Author):

Hyunyoung Yoon and Kyung Lib Jang reported in this manuscript that HBV could improve HCV replication through its HBx protein. Mechanically, HBx upregulated both the protein levels and enzyme activities of cellular DNA methyltransferase1(DNMT1) and DNMT3b, and this subsequently reduced the expression levels of the E6-associated protein (E6AP), an E3 ligase of HCV core protein, via DNA methylation. The ubiquitin-dependent proteasomal degradation of the HCV core protein was severely impaired in the presence of HBx, whereas this effect was not observed when either E6AP was ectopically expressed or restored by treatment with 5-Aza-2'dC. Although the authors have made several additional data to improve the quality of the manuscript, some issues still remain.

Major issues:

1. The authors showed that HBV infection efficacy was very high with low MOI and HBsAg could be detected even after 24 hours infection, which is amazing. The authors also only incubated HBV 1 hour with the cells and lead to that so high infection efficacy. Usually, most studies incubate the viruses for 24 hours.

→ Most HBV infection in the present study was actually performed at 30 MOI for 72 h (initial infection for 24 h, washing, and further incubation for 48 h). The protocol in the Materials and Methods (lines 443-449) was corrected accordingly. The HBV replication under the condition appears to be sufficient to show the effects of HBx on the HCV core protein and HCV replication, as demonstrated with various methods including western blot analysis, IFA, flow cytometry, qRT-PCR, and Southern blotting, *etc.* Fig. 1G is the only data obtained by coinfecting cells with HBV and HCV at 10 MOI each for 24 h (initial incubation for 1h, washing, and incubation for an additional 23 h), which was aimed to compare the effects of Lamivudine on HBV and HCV at the early stages of virus replication during coinfection. It was

possible to detect both intracellular HBV and HCV proteins and extracellular HBV and HCV particles by western blot analysis and qPCR, respectively, as shown in Fig. 1G to H. Overall, please consider that the present study attempted to provide a suitable HBV/HCV coinfection system, which allows replication of both HBV and HCV in human hepatoma cells. Therefore, it does not have to involve robust replication of HBV.

2. The authors showed in figure 1 that lamivudine treatment for 48 h can reduce HBx protein expression, which is also surprised to me. As we all known, lamivudine inhibits HBV replication via block viral reverse transcription, resulting in low mature viral particles secretion. Why lamivudine treatment could significantly reduce HBx expression only in a 48 h of viral replication period?

→ The Lamivudine experiment was performed by infecting cells for 24 h. Lamivudine may act to reduce HBx expression at the earlier time points, as the reviewer commented, but it took at least 24 h to show the difference between untreated and treated samples because the expression of HBx and HBsAg in the untreated cells started to be detectable at 24 hours postinfection, as shown in Fig. S3.

3. As the results shown, E6AP could also inhibit HBx expression. how? Does it similar to HCV core protein via the ubiquitin-dependent proteasomal degradation?

→ Yes. E6AP also induces the ubiquitin-dependent proteasomal degradation of HBx (please refer to lines 360-362)

4. The authors only showed the HBx could reduce E6AP expression to improve HCV core protein expression. Is there other pathways for HBx stimulate HBV core protein expression besides E6AP? The authors could design the shRNA or siRNA to downregulate E6AP to study if HBx could still influence on HCV replication.

→ The potential of HBx to downregulate HCV core protein levels was not detectable when the difference in E6AP levels was removed by ectopic expression, 5-Aza-2'dC treatment, DNMT1 knockdown, or E6AP knockdown (Fig. 3 and 5), suggesting that E6AP is a unique factor that is involved in the HBx-mediated upregulation of HCV core protein levels during coinfection (lines 349-353).

5. As mentioned above, E6AP also inhibit HBx expression. Both HCV core protein and HBx influence on E6AP expression. why only HCV replication was improved rather than HBV? Are there other pathways impact HBV replication?

→ It remains uncertain why the downregulation of E6AP levels by HBx and HCV core protein provides more benefits to HCV core protein rather than HBx and results in HCV dominance. The proteasomal degradation of HBx appears to be primarily regulated by another E3 ligase termed Siah-1 (47). As Siah-1 is a target of p53, HBx activates Siah-1 expression via upregulation of p53 levels, resulting in downregulation of HBx levels (47). It has also been demonstrated that HCV core protein inhibits HBV replication via activation of Siah-1 expression during coinfection (21), suggesting that Siah-1 is dominant over E6AP in the regulation of the HBx levels. It is thus interesting to investigate the relative roles of Siah-1 and E6AP in the regulation of HBx, which can affect viral dominancy during coinfection (lines 363-372).

6. Fig.1C and 1D, why HBx and HCV core protein were shown in different appearance in these two IF images?

→ Both HBx and HCV core proteins are mostly located in the cytoplasm in Fig1C and 1D (now Fig. 1E). Depending on their intensity, the IF images can be detected in different appearance.

Minor issue:

1. Line 87, HCV replication should be "HBV replication".

→ It is corrected.

2. Line 130, why the authors also use GEQ to normalize HCV? Did the author use RT-qPCR to quantify HCV RNA levels to determine HCV amounts of infection? Usually, the MOI of HCV was performed by TCID50.

→ Yes. RT-qPCR was employed in the present study to quantify HCV RNA levels and determine HCV titers as described in the M & M section (lines 457-461).

Reviewer #2 (Comments for the Author):

This study described that HBV X protein stimulated HCV replication by inhibiting the expression of E6-associated protein via DNA methylation. The decreased expression of E6-associated protein protected the HCV core protein from proteasomal degradation, which could contribute to HCV dominance during HBV/HCV coinfection. The experiments have been carried out with care. The reviewer only has 1 major comment that will need to be addressed.

Major comment

1. Since all study works on the coinfection of HBV and HCV, it would be important to show that the stimulation of HCV replication by HBV infection or expression of HBV X protein occurs in the same cells. The reviewer suggests the authors repeat the Figure 1 C and 1D and analyze the expression of HBx and HCV core by flow cytometry.

→ Based on the comment, we performed flow cytometric analysis of the mock-infected, HBV monoinfected, HCV monoinfected, and HBV/HCV coinfecting Huh7D-NTCP cells to determine the proportions of cells expressing HBx and/or HCV core protein, as shown in Fig. 1D, 1F, S2A, and S2B. As a result, we found that 80-90 % of the coinfecting cells express both HBx and HCV core protein. In addition, the average fluorescent signal from the HCV core protein was stronger in the coinfecting cells, as compared to that in the HCV monoinfected cells (Fig. 1D, 1F, S2A, and S2B). Please also refer to lines 120-125 and 140-143.

Minor comment

1. line 87, replace the "inhibits HCV replication" with "inhibits HBV replication".

→ It is corrected.

October 8, 2022

Prof. Kyung Lib Jang
Pusan National University
Microbiology
san 30, Jangjeon-dong, Keumjeong-gu
Busan 46241
Korea (South), Republic of

Re: Spectrum01432-22R1 (Hepatitis B virus X protein stimulates hepatitis C virus replication by protecting HCV core protein from E6AP-mediated proteasomal degradation)

Dear Prof. Kyung Lib Jang:

Link Not Available

Sincerely,

Leiliang Zhang

Journals Department
Reviewer comments:

Reviewer #1 (Comments for the Author):

In this regard, the reviewed manuscript the authors have included the additional experiments to the issues and discussed it in depth, which is now provided in the new version of the manuscript.

Overall the quality of the manuscript has substantially improved.

Reviewer #2 (Comments for the Author):

Thanks for performing the flow experiments in Fig. S2. I have a following question about data display. Instead of showing the single color by histograms, it would be better to show the data by dot plots in the two-dimensional display (X axis: HBx-FITC, Y axis: HCV core-647). In this way, the expression of HBx and HCV core can be shown in the single cell, which makes the data more convincing.

Staff Comments:

Preparing Revision Guidelines

Please return the manuscript within 60 days; if you cannot complete the modification within this time period, please contact me. If you do not wish to modify the manuscript and prefer to submit it to another journal, please notify me of your decision immediately so that the manuscript may be formally withdrawn from consideration by Microbiology Spectrum.

In this regard, the reviewed manuscript the authors have included the additional experiments to the issues and discussed it in depth, which is now provided in the new version of the manuscript.

Overall the quality of the manuscript has substantially improved.

Reviewer #1 (Comments for the Author):

Hyunyoung Yoon and Kyung Lib Jang reported in this manuscript that HBV could improve HCV replication through its HBx protein. Mechanically, HBx upregulated both the protein levels and enzyme activities of cellular DNA methyltransferase1(DNMT1) and DNMT3b, and this subsequently reduced the expression levels of the E6-associated protein (E6AP), an E3 ligase of HCV core protein, via DNA methylation. The ubiquitin-dependent proteasomal degradation of the HCV core protein was severely impaired in the presence of HBx, whereas this effect was not observed when either E6AP was ectopically expressed or restored by treatment with 5-Aza-2'dC. Although the authors have made several additional data to improve the quality of the manuscript, some issues still remain.

Major issues:

1. The authors showed that HBV infection efficacy was very high with low MOI and HBsAg could be detected even after 24 hours infection, which is amazing. The authors also only incubated HBV 1 hour with the cells and lead to that so high infection efficacy. Usually, most studies incubate the viruses for 24 hours.

→ Most HBV infection in the present study was actually performed at 30 MOI for 72 h (initial infection for 24 h, washing, and further incubation for 48 h). The protocol in the Materials and Methods (lines 443-449) was corrected accordingly. The HBV replication under the condition appears to be sufficient to show the effects of HBx on the HCV core protein and HCV replication, as demonstrated with various methods including western blot analysis, IFA, flow cytometry, qRT-PCR, and Southern blotting, *etc.* Fig. 1G is the only data obtained by coinfecting cells with HBV and HCV at 10 MOI each for 24 h (initial incubation for 1h, washing, and incubation for an additional 23 h), which was aimed to compare the effects of Lamivudine on HBV and HCV at the early stages of virus replication during

coinfection. It was possible to detect both intracellular HBV and HCV proteins and extracellular HBV and HCV particles by western blot analysis and qPCR, respectively, as shown in Fig. 1G to H. Overall, please consider that the present study attempted to provide a suitable HBV/HCV coinfection system, which allows replication of both HBV and HCV in human hepatoma cells. Therefore, it does not have to involve robust replication of HBV.

2. The authors showed in figure 1 that lamivudine treatment for 48 h can reduce HBx protein expression, which is also surprised to me. As we all known, lamivudine inhibits HBV replication via block viral reverse transcription, resulting in low mature viral particles secretion. Why lamivudine treatment could significantly reduce HBx expression only in a 48 h of viral replication period?

→ The Lamivudine experiment was performed by infecting cells for 24 h. Lamivudine may act to reduce HBx expression at the earlier time points, as the reviewer commented, but it took at least 24 h to show the difference between untreated and treated samples because the expression of HBx and HBsAg in the untreated cells started to be detectable at 24 hours postinfection, as shown in Fig. S3.

3. As the results shown, E6AP could also inhibit HBx expression. how? Does it similar to HCV core protein via the ubiquitin-dependent proteasomal degradation?

→ Yes. E6AP also induces the ubiquitin-dependent proteasomal degradation of HBx (please refer to lines 360-362)

4. The authors only showed the HBx could reduce E6AP expression to improve HCV core protein expression. Is there other pathways for HBx stimulate HBV core protein expression besides E6AP? The authors could design the shRNA or siRNA to downregulate E6AP to study if HBx could still influence on HCV replication.

→ The potential of HBx to downregulate HCV core protein levels was not detectable when the difference in E6AP levels was removed by ectopic expression, 5-Aza-2'dC treatment, DNMT1 knockdown, or E6AP knockdown (Fig. 3 and 5), suggesting that E6AP is a unique factor that is involved in the HBx-mediated upregulation of HCV core protein levels during coinfection (lines 349-353).

5. As mentioned above, E6AP also inhibit HBx expression. Both HCV core protein and HBx influence on E6AP expression. why only HCV replication was improved rather than HBV? Are there other pathways impact HBV replication?

→ It remains uncertain why the downregulation of E6AP levels by HBx and HCV core protein provides more benefits to HCV core protein rather than HBx and results in HCV dominance. The proteasomal degradation of HBx appears to be primarily regulated by another E3 ligase termed Siah-1 (47). As Siah-1 is a target of p53, HBx activates Siah-1 expression via upregulation of p53 levels, resulting in downregulation of HBx levels (47). It has also been demonstrated that HCV core protein inhibits HBV replication via activation of Siah-1 expression during coinfection (21), suggesting that Siah-1 is dominant over E6AP in the regulation of the HBx levels. It is thus interesting to investigate the relative roles of Siah-1 and E6AP in the regulation of HBx, which can affect viral dominancy during coinfection (lines 363-372).

6. Fig.1C and 1D, why HBx and HCV core protein were shown in different appearance in these two IF images?

→ Both HBx and HCV core proteins are mostly located in the cytoplasm in Fig1C and 1D (now Fig. 1E). Depending on their intensity, the IF images can be detected in different appearance.

Minor issue:

1. Line 87, HCV replication should be "HBV replication".

→ It is corrected.

2. Line 130, why the authors also use GEQ to normalize HCV? Did the author use RT-qPCR to quantify HCV RNA levels to determine HCV amounts of infection? Usually, the MOI of HCV was performed by TCID50.

→ Yes. RT-qPCR was employed in the present study to quantify HCV RNA levels and determine HCV titers as described in the M & M section (lines 457-461).

Reviewer #2 (Comments for the Author):

This study described that HBV X protein stimulated HCV replication by inhibiting the expression of E6-associated protein via DNA methylation. The decreased expression of E6-associated protein protected the HCV core protein from proteasomal degradation, which could contribute to HCV dominance during HBV/HCV coinfection. The experiments have been carried out with care. The reviewer only has 1 major comment that will need to be

addressed.

Major comment

1. Since all study works on the coinfection of HBV and HCV, it would be important to show that the stimulation of HCV replication by HBV infection or expression of HBV X protein occurs in the same cells. The reviewer suggests the authors repeat the Figure 1 C and 1D and analyze the expression of HBx and HCV core by flow cytometry.

→ Based on the comment, we performed flow cytometric analysis of the mock-infected, HBV monoinfected, HCV monoinfected, and HBV/HCV coinfecting Huh7D-NTCP cells to determine the proportions of cells expressing HBx and/or HCV core protein, as shown in Fig. 1D, 1F, S2A, and S2B. As a result, we found that 80-90 % of the coinfecting cells express both HBx and HCV core protein. In addition, the average fluorescent signal from the HCV core protein was stronger in the coinfecting cells, as compared to that in the HCV monoinfected cells (Fig. 1D, 1F, S2A, and S2B). Please also refer to lines 120-125 and 140-143.

Minor comment

1. line 87, replace the "inhibits HCV replication" with "inhibits HBV replication".

→ It is corrected.

October 15, 2022

Prof. Kyung Lib Jang
Pusan National University
Microbiology
san 30, Jangjeon-dong, Keumjeong-gu
Busan 46241
Korea (South), Republic of

Re: Spectrum01432-22R2 (Hepatitis B virus X protein stimulates hepatitis C virus replication by protecting HCV core protein from E6AP-mediated proteasomal degradation)

Dear Prof. Kyung Lib Jang:

Your manuscript has been accepted, and I am forwarding it to the ASM Journals Department for publication. You will be notified when your proofs are ready to be viewed.

Sincerely,

Leiliang Zhang
Editor, Microbiology Spectrum
